# Compatibility of Entomopathogenic Nematodes with Chemical Insecticides for the Control of *Drosophila suzukii* (Diptera: Drosophilidae)

**DOI:** 10.3390/plants13050632

**Published:** 2024-02-25

**Authors:** Sérgio da Costa Dias, Andressa Lima de Brida, Maguintontz Cedney Jean-Baptiste, Luís Garrigós Leite, Sergio M. Ovruski, Jana C. Lee, Flávio Roberto Mello Garcia

**Affiliations:** 1Department of Ecology, Zoology and Genetics, Institute of Biology, Federal University of Pelotas, Pelotas 96010-900, RS, Brazil; sergiodacoxta@gmail.com (S.d.C.D.); andressa_brida23@hotmail.com (A.L.d.B.); magcedneyjeanbaptiste@yahoo.fr (M.C.J.-B.); 2Centro Experimental de Campinas, Instituto Bilógico, Rod. Heitor Penteado km 3, Campinas 13001-970, SP, Brazil; lgleite@biologico.sp.gov.br; 3IEMEN, Biological Pest Control Division, PROIMI Biotechnology, CCT NOA Sur-CONICET, Belgrano y Pje, Caseros Aveniew, San Miguel de Tucumán T4001MVB, Tucumán, Argentina; sovruski@conicet.gov.ar; 4Horticultural Crops Disease and Pest Management Research Unit, USDA-ARS, 3420 NW Orchard Ave., Corvallis, OR 97330-5014, USA; jana.lee@usda.gov

**Keywords:** biological control, spotted-wing drosophila, virulence, *Heterorhabditis*, *Steinernema*

## Abstract

The spotted-wing drosophila, *Drosophila suzukii* (Matsumura) (Diptera: Drosophilidae), is a pest that reduces the productivity of small fruits. Entomopathogenic nematodes (EPNs) and chemical insecticides can suppress this pest, but the compatibility of the two approaches together requires further examination. This laboratory study evaluated the compatibility of *Steinernema brazilense* IBCBn 06, *S. carpocapsae* IBCBn 02, *Heterorhabditis amazonensis* IBCBn 24, and *H. bacteriophora* HB with ten chemical insecticides registered for managing *D. suzukii* pupae. In the first study, most insecticides at the recommended rate did not reduce the viability (% of living infective juveniles (IJs)) of *S. braziliense* and both *Heterorhabditis* species. The viability of *S. carpocapsae* was lowered by exposure to spinetoram, malathion, abamectin, azadirachtin, deltamethrin, lambda-cyhalothrin, malathion, and spinetoram after 48 h. During infectivity bioassays, phosmet was compatible with all the EPNs, causing minimal changes in infectivity (% pupal mortality) and efficiency relative to EPN-only controls, whereas lambda-cyhalothrin generally reduced infectivity of EPNs on *D. suzukii* pupae the most, with a 53, 75, 57, and 13% reduction in infectivity efficiency among *H. bacteriophora, H. amazonensis, S. carpocapsae*, and *S. brazilense*, respectively. The second study compared pupal mortality caused by the two most compatible nematode species and five insecticides in various combinations. Both *Heterorhabditis* species caused 78–79% mortality among *D. suzukii* pupae when used alone, and were tested in combination with spinetoram, malathion, azadirachtin, phosmet, or novaluron at a one-quarter rate. Notably, *H. bacteriophora* caused 79% mortality on *D. suzukii* pupae when used alone, and 89% mortality when combined with spinetoram, showing an additive effect. Novaluron drastically reduced the number of progeny IJs when combined with *H. amazonensis* by 270 IJs and *H. bacteriophora* by 218. Any adult flies that emerged from EPN–insecticide-treated pupae had a shorter lifespan than from untreated pupae. The combined use of *Heterorhabditis* and compatible chemical insecticides was promising, except for novaluron.

## 1. Introduction 

The spotted-wing drosophila, *Drosophila suzukii* (Matsumura) (Diptera: Drosophilidae), is a fruit fly native to Asia that is currently found in North and South America, Europe, Africa, and Oceania. It is a polyphagous quarantine pest with high economic importance due to its ability to infest a variety of fruits. Unlike other drosophilids, which generally lay eggs on damaged or decomposing fruits, *D. suzukii* can lay eggs inside healthy fruits [1]. Primary damage is caused by the larvae consuming the pulp and softening the fruit. Secondary damage is caused by the entry of phytopathogenic microorganisms once the fruit has been punctured [2]. 

Though chemical insecticides are effective [3], they may kill non-target species, pollute the environment, lead to insecticide-resistant pest populations, and harm human health [4]. Therefore, biological control using entomopathogenic nematodes (EPNs) is a promising alternative [5] given their efficiency, adaptive capacity, and easy application. Furthermore, nematodes search in the soil for the host through chemoreceptor mechanisms and can be selective for the target insect species [6]. 

EPNs are often applied with other phytosanitary products (chemical, natural, and biological), fertilizers, and soil correctives, and can be mixed in tanks [7]. For example, *Steinernema carpocapsae* (Weiser), *S. feltiae* Filipjev, and *Heterorhabditis bacteriophora* (Poinar) can survive when exposed to various types of chemical pesticides [8]. The action of pesticides on entomopathogenic organisms varies according to the species and lineage of the pathogens, as well as the chemical nature and concentrations of the products used [9]. The effects of pesticides on EPNs can be evaluated by (1) observing the viability and behavior of infective juveniles (IJs) exposed to various concentrations of a given pesticide for different periods and (2) observing the ability of IJs to infect host insects [10]. The compatibility of EPNs with brief exposures to chemical insecticides is an important factor in successful integrated pest management (IPM) [11]. 

The objective of this study was to evaluate the compatibility of *Steinernema brazilense* IBCBn 06 (isolate designation), *S. carpocapsae* IBCBn 02, *Heterorhabditis amazonensis* IBCBn 24, and *H. bacteriophora* HB with different chemical insecticides for the control of *D. suzukii* pupae, and to evaluate the longevity of surviving adult flies. *Steinernema carpocapsae* and *H. bacteriophora* were selected since they are commercially available and laboratory trials with these species have been promising in multiple countries [5]. *Steinernema braziliense* and *H. amazonensis* were selected since they are important native nematodes in Brazil. These two Brazilian isolates have shown promise in controlling fruit flies.

## 2. Results 

### 2.1. Study 1—Compatibility of EPNs with Chemical Insecticides

Insecticides can be incompatible by reducing the viability of IJs and/or infectivity rates. For *S. carpocapsae* IBCBn 02, deltamethrin, spinetoram, malathion, abamectin, azadirachtin, and lambda-cyhalothrin reduced the viability of the IJs relative to the nematode-only control treatment by 55.1%, 35.5%, 54.8%, 77.30%, 53.26%, and 40.16%, respectively (100%—% viability with insecticide, Table 1). Likewise, deltamethrin, spinetoram, malathion, and lambda-cyhalothrin also reduced infectivity relative to the controls (Table 2) and were considered toxic by IOBC standards since the reduction in infectivity efficiency (∆E%) exceeded 30% [12]. For *S. brazilense* IBCBn 06, only lambda-cyhalothrin significantly reduced IJ viability by 12.6% (Table 1) but was not classified as toxic since ∆E% was 12.5% (Table 2). Overall, *S. brazilense* had low infectivity rates, causing 6–16% pupal mortality which was lower than the 10–42% rates seen with the other isolates (Table 2). Furthermore, *S. braziliense* was only negatively affected by thiamethoxam and acetamiprid, which were slightly toxic to the nematodes lowering infectivity efficiency by 38 and 63%, respectively (∆E% in Table 2).

For *H. amazonensis* IBCBn 24, only abamectin reduced the viability of IJs by 15.5% relative to the nematode-only control (Table 1). *Heterhabditis amazonensis* was highly affected by nine insecticides regarding infectivity; only phosmet did not affect the nematodes’ infectivity, with a 0% change in efficiency. For *H. bacteriophora*, only malathion and abamectin significantly lowered IJ viability by 12.64% and 11.71% relative to the control, respectively (Table 1). Five insecticides affected infectivity, as *H. bacteriophora* infectivity was lowered with spinetoram, abamectin, azadirachtin, novaluron, and lambda-cyhalothrin, and efficiency was lowered by 32, 37, 47, 53, and 53%, respectively (∆E% in Table 2).

Generally, *H. amazonensis* and *H. bacteriophora* showed more compatibility with the insecticides. In comparisons across the four species, *H. bacteriophora* exhibited higher infectivity rates with all ten of the insecticides, whereas *S. braziliense* exhibited lower infectivity with nine insecticides (see comparison using capital letters in Table 2). *Steinernema carpocasae* had the lowest viability with all ten insecticides (see comparison using capital letters in Table 1). Thus, the nematodes with lower viability and infectivity responses were not tested in the second study.

### 2.2. Study 2—Effectiveness of Selected EPNs + Insecticides

The isolates *H. amazonensis* IBCBn 24 and *H. bacteriophora* HB and spinetoram, malathion, azadirachtin, phosmet, and novaluron, either separately or combined, caused 7–95% pupal mortality in *D. suzukii* (Table 3). Mortality caused by the EPN + insecticide combinations was significantly higher than the negative control of water (Table 3). A combination of *H. amazonensis* + spinetoram resulted in the greatest mortality of *D. suzukii* pupae at 95%, with a significant 17.5% increase from the EPN alone with 77.5% mortality. *Heterorhabditis bacteriophora* caused 78.75% mortality of the pupae when used alone, and 88.75% mortality with spinetoram combined, causing a 10% numerical increase (Table 3).

The addition of all the tested insecticides reduced the number of IJs developing in the treated pupae (Table 3). Novaluron caused the most drastic reduction with a 270 IJ/pupa reduction when combined with *H. amazonensis*, and 218 IJ/pupa reduction with *H. bacteriophora.* While novaluron did not reduce *D. suzukii* pupal mortality when combined with either *Heterorhabditis* species compared to the EPN-only treatments, novaluron negatively affected the developing IJs in *D. suzukii* pupae (Table 3). The other four insecticides reduced nematode production by 114–210 IJ/pupa with *H. amazonensis*, and by 48–172 IJ/pupa with *H. bacteriophora*. 

Longevity was shortened among the surviving adult *D. suzukii* from all the treatments with EPN and/or insecticides compared to the untreated control pupae (F = 41.94; d.f = 17, 126; *p* ˂ 0.0001) (Figure 1). The surviving adults lived ~3.36 days less when exposed to both EPN and insecticides than insecticide alone. Also, the adults lived ~5 days or less when exposed to *H. amazonensis +* azadirachtin, *H. amazonensis* + phosmet, *H. bacteriophora* + spinetoram, *H. bacteriophora +* malathion, and *H. bacteriophora* + phosmet. 

## 3. Discussion 

The isolates *H. bacteriophora* HB, *H. amazonensis* IBCBn 24, *S. brazilense* IBCBn 06, and *S. carpocapsae* IBCBn 02 were often compatible with the ten chemical insecticides tested, as 48 h of exposure resulted in no significant reduction in the viability of IJs in 31 out of the 40 EPN–insecticide combinations tested. Of the four species, *S. carpocaspsae* experienced the most reduction in viability with six out of ten insecticides. In contrast, the pesticides applied directly to *S. carpocapsae* of all the strains at the recommended dose did not affect viability after 3 h of exposure [13]. The different outcomes between the studies may be due to exposure durations of 48 versus 3 h. Other studies showed that the IJs of other *Steinernema* spp. were tolerant to insecticides. Botanical and chemical insecticides at the recommended doses did not affect the viability of *S. feltiae* 72 h after exposure [14], nor did phosmet, fipronil, and thiamethoxam affect the viability of *Steinernema* sp. [15]. On the other hand, lambda-cyhalothrin affected the viability of *S. carpocapsae* and *S. amazonensis* in this study, which corroborates Negrisoli Jr. et al.’s [16] study with *S. carpocapsae* and *S. glaseri* (Steiner). 

In addition to minimal changes in IJ viability, a compatible insecticide should not reduce the subsequent infectivity rate of EPNs. Though IJs may remain alive, an insecticide can still reduce infectivity rates by hampering the nematode’s dispersal ability and attraction to the host [17]. Phosmet was the most compatible out of the ten insecticides tested and did not reduce the infectivity of the four nematode species. Abamectin was somewhat incompatible as it reduced the infectivity of *S. braziliense* and *H. amazonensis* but not *H. bacteriophora* or *S. carpocapsae* in this study. Koppenhöfer et al. [18] and Kary et al. [6] observed that *S. feltiae* was negatively affected by abamectin, while the effect on *H. bacteriophora* was very slight. Since the thickness of the epicuticle, cortical, and median cuticle layers of IJs differs between species [19], the different susceptibilities to abamectin between species may be due to differences in cuticles. Abamectin may damage the juvenile cuticle by affecting its permeability and loss of annulations and grooves in the body [6]. The harmful effects of abamectin can lower the viability and infectivity of IJs [20].

Next, thiamethoxam was more toxic among the insecticides tested: it reduced the infectivity of three out of the four nematode species. Thiamethoxam is a widely used systemic insecticide in orchards worldwide, especially in Brazil, for psyllid, sharpshooter, mealybug, aphid, and leafminer control in fruit orchards. Thiamethoxam is applied by soil-drench, where *D. suzukii* often pupate [21]. Hence, its application to soil combined with EPN applications might compromise their persistence in agroecosystems [21]. Lastly, lambda-cyhalothrin was the most toxic of the tested insecticides; it reduced infectivity for all four EPN species. Likewise, negative results were obtained for the EPNs after being exposed to lambda-cyhalothrin [22]. 

Our study supports integrating both forms of protection into agronomic practices. A one-quarter dose of spinetoram provided an additive effect when combined with *H. amazonensis* by increasing *D. suzukii* pupal mortality by 17.5%. Likewise, Kary et al. [6] reported that *H. bacteriophora* and *S. feltiae* were effective control agents against the tuber moth when used with abamectin, providing an increase of 17% in protection. Also, Navarro et al. [23] reported that imidacloprid worked additively with *H. sonorensis*, and dinotefuran worked additively with *S. riobrave* and *H. sonorensis*. For the other EPN–insecticide combinations in Study 2, there were no additive effects nor negative effects. This is similar to the conclusions of Polavarapu et al. [24], who found that *Steinernema scarabaei* and *H. bacteriophora* in combination with thiamethoxam and phosmet against scarab did not have an additive effect nor a negative effect. While our study focused on integrating EPNs and insecticides, other chemicals that EPNs encounter in the field require consideration. For example, *H. bacteriophora* was found to maintain high infectivity in *G. mellonella* caterpillars when exposed to the fungicides mancozeb and metalaxyl + folpet [24,25,26].

The compatibility obtained by combining EPNs and insecticides can be caused by the chemical ingredient stressing the insect, affecting its physiology and humoral defense, and consequently making it more susceptible to infections by nematodes [27]. Also, the increased efficacy of EPN–insecticide combinations may also be due to the nematodes’ increased movement and nictation activity after exposure to an insecticide [28]. Lastly, Gaugler et al. [29] observed that the compatibility of *H. bacteriophora* + phosmet on the mortality of scarabeid larvae *Cyclocephala* sp. (Coleoptera: Scarabaeidae) was caused by changes in the insect’s behavior prompted by the insecticide. Scarab larvae ceased to clean their cuticle or mandibles and did not remove nematodes and other natural enemies in the process. 

In this study, the longevity of *D. suzukii* adults was shortened by the presence of IJs of *H. bacteriophora* HB, and *H. amazonensis* IBCBn 24, either applied alone or in combination with insecticides. After penetrating the insect’s integument, IJs usually cause mortality between 24 and 48 h. The emergence of infected adults indicates resistance to infection during the pupal period [30].

## 4. Materials and Methods

Experiments were performed in the Insect Ecology Laboratory of the Federal University of Pelotas, in the state of Rio Grande Sul, Brazil. *Drosophila suzukii* were reared by placing adults in flat-bottomed glass containers (85 mm high × 25 mm in diameter ± 0.5 mm) in a climate chamber (ELETROLab^®^, model EL 212, São Paulo, Brazil) at 25 ± 1 °C, 70 ± 10% RH, and a 12 h:12 h L:D photoperiod. Adults were given a diet that consisted of 500 mL of water, agar (4 g), yeast (20 g), corn flour (40 g), sugar (50 g), 1.5 mL of propionic acid, and Nipagin (10%; 3.5 mL) [31] for food as well as an egg laying substrate. Pupae less than 24 h old were used in the experiments. EPNs were obtained from the ‘Oldemar Cardim Abreu’ Entomopathogenic Nematode Bank of the Biology Institute of São Paulo. The isolates *S. brazilense* IBCBn 06, *S. carpocapsae* IBCBn 02, *H. amazonensis* IBCBn 24, and *H. bacteriophora* HB were multiplied separately in caterpillars of the fourth and fifth instars *Galleria mellonella* L. (Lepidoptera: Pyralidae) [32]. The infective juveniles (IJs) used in the experiments were within six days of emergence.

### 4.1. Chemical Insecticides

We used 10 insecticides from different chemical groups, prepared at the concentration recommended by the vendor for use in strawberry crops; this fruit is attacked the most by *D. suzukii* in Brazil (Table 4). Based on this concentration, 350 mL solutions of each product were prepared. The insecticides were chosen based on their availability in the market and reported efficacy for the control of *D. suzukii* in Brazil. 

### 4.2. Study 1—Compatibility of EPNs with Chemical Insecticides 

The compatibility of *S. brazilense*, *S. carpocapsae*, *H. bacteriophora*, or *H amazonesis* with the chemical insecticides was evaluated by following Negrisoli Jr. et al. [10]. First, 1000 mL of each insecticide solution was prepared in water (Table 4). Then, 1 mL of the insecticide solution was placed in an 8 mL glass test tube (2.5 cm diam. × 8 cm high), followed by the addition of 2500 IJs in 1 mL of distilled water. Each insecticide–nematode treatment combination was replicated in five tubes. The tubes were agitated and maintained in a climate chamber at 22 ± 1 °C and RH of 70 ± 10%. First, nematode ‘viability’ was evaluated 48 h after exposure. An aliquot of 0.1 mL of the suspension was removed from each tube and approximately 100 IJs were observed with a stereomicroscope at 40×. The IJs were considered dead when they did not respond to the stimulus of a stylus.

The ‘infectivity’ of the nematodes was also tested in the same replicates set up for measuring viability. To wash off the insecticides, the tubes were filled with 3 mL of distilled water, and the solutions were left to settle for 30 min in a refrigerator. About 3 mL of the supernatant was decanted and the remaining substance was washed three times with distilled water. After the last washing, 0.2 mL with ~100 IJs was pipetted into the bottom of a Petri dish (90 diam. × 15 mm), where 10 24-h-old *D. suzukii* pupae were added. The dishes were then placed in a climate chamber at 22 ± 1 °C and RH of 70 ± 10% for five days, after which pupal mortality was recorded. ‘Infectivity’ is the percentage of dead pupae.

The reduction in efficiency ‘∆E%’ reflects whether nematodes infect *D. suzukii* pupae less when nematodes were previously exposed to insecticides. ∆E% is calculated by the following: ∆E% = (1 − mt/mc) × 100, where mt is the pupal mortality of the treatment and mc is the mortality of the control [33], based on guidelines from the International Organization for Biological and Integrated Control (IOBC). ∆E% values were classified according to IOBC/WPRS [12] as follows: 1—nontoxic insecticides that reduce infectivity efficiency by less than 30%, 2—slightly toxic (30–79%), 3—moderately toxic (80–99%), and 4—toxic (>99%). Insecticides compatible with EPNs should cause high pupal mortality (infectivity) and cause minimal changes in infectivity efficiency (∆E%). 

### 4.3. Study 2—Effectiveness of Selected EPNs + Insecticides 

Based on Study 1, we continued trials with the two most compatible nematode species, *H. bacteriophora* HB and *H. amazonensis* IBCBn 24, in combination with either spinetoram, azadirachtin, malathion, phosmet, or novaluron at ¼ of the recommended dose for strawberries (Table 4). The sub-dose of ¼ was used to reduce the environmental impacts of the insecticides and to observe the potential compatibility of the combinations with nematodes; otherwise, a full dose of an insecticide alone may already kill most *D. suzukii* pupae, masking any positive impacts of EPN combinations. Concentrations of 5400 IJs for *H. bacteriophora* and 1800 IJs for *H. amazonensis* were used since these concentrations caused the greatest mortality in *D. suzukii* pupae in Study 1. 

Treatments comprised each nematode and each insecticide either alone or in combination and were as follows: (1) *H. amazonensis (H. a.)*; (2) *H. bacteriophora (H. b.)*; (3) spinetoram; (4) azadirachtin; (5) malathion; (6) phosmet; (7) novaluron; (8) *H.a.* + spinetoram; (9) *H.a.* + azadirachtin; (10) *H. a.* + malathion; (11) *H. a.* + phosmet; (12) *H. a* + novaluron; (13) *H. b.* + spinetoram; (14) *H. b.* + azadirachtin; (15) *H. b.* + malathion; (16) *H. b.* + phosmet; (17) *H. b.* + novaluron, and (18) water control. Each treatment had eight replications, each consisting of 10 pupae grouped in a 50 mL plastic jar, filled with 50 g of sterilized sand with 10% moisture by weight. 

The 1 mL nematode suspension or water (no EPN) was mixed with 3 mL of distilled water in a vial. Then, 1 mL of insecticide or water was added and the vial shaken; then, all 5 mL was pipetted into each plastic jar containing pupae. As a negative control, only sterile water was inoculated, and as a positive control, only the insecticide solution was used without nematodes. The jars were incubated at 22 ± 1 °C and 70 ± 10% RH for six days, after which we recorded the number of dead pupae. In the treatments with nematodes, the dead pupae were dissected to count the IJs. 

The surviving adult *D. suzukii* that emerged from the pupal treatment were placed in individual 300 mL plastic cups and observed for longevity. The cups had a 5 cm diameter hole in the lid covered with voile fabric to allow air circulation. The adults were fed 10 g of artificial diet and 1 mL of distilled water in a cotton wick. The flies were incubated at 22 ± 1 °C and 70 ± 10% RH until death. 

### 4.4. Statistical Analysis

A generalized linear model (GLM) with an appropriate distribution analyzed the treatment effects on the viability %, and infectivity % of the EPNs in Study 1. The treatments were compared in two manners: differences between the insecticides for a given nematode species, and differences between the nematode species with a given insecticide. In Study 2, the pupal mortality %, number of IJs that emerged per *D. suzukii* pupa, and longevity of the surviving *D. suzukii* adults were compared in a GLM. The goodness of fit of the data to the model was assessed by using a half-normal probability plot with a simulated envelope [34]. When significant differences between the treatments were detected, multiple comparisons (Tukey HSD test, *p* < 0.05) were performed using the glht function of the Multicomp package, with adjustment of *p*-values. These analyses were performed in R software version 4.2.3 [35].

## 5. Conclusions

In summary, many of the nematode–insecticide combinations tested resulted in viable IJs, with high infectivity rates, particularly among *H. amazonensis* IBCBn 24 and *H. bacteriophora* HB. Further testing showed that the combined use of the EPNs and compatible chemical insecticides had neutral or additive effects, except for novaluron, which negatively affected EPN propagation within the treated *D. suzukii* pupae. The use of some Brazilian isolates of EPNs with insecticides is promising against *D. suzukii* within an integrated pest management approach.

## Figures and Tables

**Figure 1 plants-13-00632-f001:**
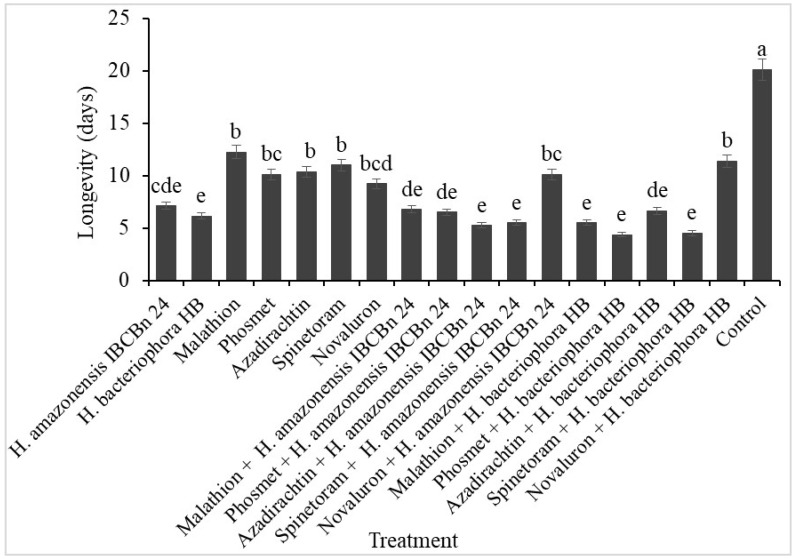
Longevity of *Drosophila suzukii* adults exposed to a combination of insecticides and *H. amazonensis* IBCBn 24 or *H. bacteriophora* HB. Different letters denote differences identified by the Tukey test, *p* < 0.05.

**Table 1 plants-13-00632-t001:** Viability (mean % living IJs ± SE) of *S. brazilense* IBCBn 06, *S. carpocapsae* IBCBn 02, *H. amazonensis* IBCBn 24, and *H. bacteriophora* HB after 48 h of exposure to insecticides.

Treatment	*S. brazilense* ^a^	*S. carpocapsae* ^a^	*H. amazonensis* ^a^	*H. bacteriophora* ^a^
EPN only Control	100.00 ± 0.00 aA	100.00 ± 0.00 aA	100.00 ± 0.00 aA	100.00 ± 0.00 aA
Deltametrin	93.99 ± 0.55 aB	44.90 ± 0.68 bC	93.10 ± 0.46 aB	97.34 ± 0.67 aA
Lambda-cyalothrin	87.84 ± 0.75 bC	59.84 ± 1.46 bD	94.20 ± 1.65 aA	90.60 ± 2.40 aB
Malathion	96.69 ± 0.32 aA	45.92 ± 2.77 bC	90.72 ± 2.56 aB	87.36 ± 1.19 bB
Phosmet	94.36 ± 1.06 aA	84.20 ± 1.35 aB	97.64 ± 1.26 aA	94.92 ± 0.87 aA
Azadirachtin	95.30 ± 1.28 aA	46.74 ± 8.86 bC	98.89 ± 0.23 aA	91.69 ± 0.66 aB
Thiamethoxam	97.34 ± 0.63 aA	92.00 ± 1.09 aB	93.48 ± 0.85 aB	97.04 ± 0.31 aA
Acetamiprid	96.84 ± 0.67 aAB	86.83 ± 0.88 aC	97.46 ± 0.69 aA	94.27 ± 0.37 aB
Spinetoram	96.69 ± 0.48 aA	64.48 ± 0.96 bB	98.24 ± 0.39 aA	96.18 ± 1.35 aA
Abamectin	95.71 ± 0.48 aA	22.70 ± 0.53 cD	84.50 ± 0.54 bC	88.29 ± 0.58 bB
Novaluron	96.24 ± 0.39 aA	88.00 ± 2.64 aB	97.76 ± 0.34 aA	95.26 ± 0.40 aA
F	32.24	67.71	17.48	30.21
d.f	10, 44	10, 44	10, 44	10, 44
*p*	<0.0001	<0.0001	<0.0001	<0.0001

^a^ Means followed by different lowercase letters in the same column and uppercase letters in the same row indicate significant differences between insecticides for a given nematod, and between nematode for a given insecticide, respectively (Tukey HSD test, *p* < 0.05).

**Table 2 plants-13-00632-t002:** Infectivity (mean % pupal mortality ± SE), ∆E% and class of *S. brazilense* IBCBn 06, *S. carpocapsae* IBCBn 02, *H. amazonensis* IBCBn 24, and *H. bacteriophora* HB after 48 h of exposure to insecticides (IOBC/WPRS protocol—15).

Treatment	*S. brazilense* IBCBn 06			*S. carpocapsae* IBCBn 02		
	Infectivity (%) ^a^	∆E% ^1^	Class/IOBC ^2^	Infectivity (%) ^a^	∆E% ^1^	Class/IOBC ^2^
EPN only Control	16.00 ± 0.70 aB	_	_	42.00 ± 1.37 aA	_	_
Deltametrin	14.00 ± 1.67 aC	12.50	1	26.00 ± 1.22 bB	38.10	2
Lambda-cyalothrin	14.00 ± 0.44 bA	12.50	1	18.00 ± 2.62 bA	57.14	2
Malathion	12.00 ± 0.70 aC	25.00	1	26.10 ± 2.09 bB	38.10	2
Phosmet	14.00 ± 0.44 aC	12.50	1	40.00 ± 2.44 aAB	4.76	1
Azadirachtin	16.00 ± 0.31 aB	0.00	1	28.00 ± 3.74 bA	33.33	2
Thiamethoxam	10.00 ± 0.94 bB	37.50	2	26.00 ± 6.78 bA	38.10	2
Acetamiprid	6.00 ± 0.70 cB	62.50	2	40.00 ± 2.72 aA	4.76	1
Spinetoram	14.00 ± 0.54 aC	12.50	1	32.00 ± 1.44 bA	23.81	1
Abamectin	14.00 ± 0.89 aC	12.50	1	36.00 ± 1.70 aA	14.29	1
Novaluron	14.00 ± 0.70 aB	12.50	1	38.00 ± 3.39 aA	9.52	1
F	12.00			5.64		
d.f	10, 44			10, 44		
*p*	<0.0001			<0.0001		
Treatment	*H. amazonensis* IBCBn 24			*H. bacteriophora* HB		
	Infectivity (%) ^a^	∆E% ^1^	Class/IOBC ^2^	Infectivity (%) ^a^	∆E% ^1^	Class/IOBC ^2^
EPN only Control	40.00 ± 1.97 aA	_	_	38.00 ± 3.74 aA	_	_
Deltametrin	14.00 ± 1.37 cC	65.00	2	34.00 ± 2.45 aA	10.52	1
Lambda-cyalothrin	10.00 ± 3.16 cA	75.00	2	18.00 ± 3.74 bA	52.63	2
Malathion	20.00 ± 1.70 bB	50.00	2	36.00 ± 2.44 aA	5.26	1
Phosmet	40.00 ± 0.00 aBC	0.00	1	38.00 ± 2.00 aA	0.00	1
Azadirachtin	24.00 ± 1.81 bAB	40.00	2	20.00 ± 3.16 bAB	47.37	2
Thiamethoxam	10.00 ± 0.63 cB	75.00	2	36.00 ± 2.44 aA	5.26	1
Acetamiprid	14.00 ± 2.44 cB	65.00	2	36.00 ± 2.44 aA	5.26	1
Spinetoram	20.00 ± 1.70 bBC	50.00	2	26.00 ± 4.00 bAB	31.58	2
Abamectin	16.00 ± 1.37 cC	60.00	2	24.00 ± 2.45 bB	36.84	2
Novaluron	24.00 ± 2.44 bB	40.00	2	18.00 ± 3.74 bB	52.63	2
F	9.16			6.60		
d.f.	10, 44			10, 44		
*p*	<0.0001			<0.0001		

^a^ Means followed by different lowercase letters in the same column and uppercase letters in the same line indicate significant differences between insecticides for a given nematode, and between nematodes for a given insecticide, respectively (Tukey HSD test, *p* < 0.05). ^1^ Change in infectivity efficiency calculated by the formula ∆E% = 100 − (1 − mt/mc) × 100, ^2^ WPRS class: 1—nontoxic (∆E% < 30%), 2—slightly toxic (∆E% = 30% to 79%), 3—moderately toxic (∆E% = 80% to 99%), and 4—toxic (∆E% > 99%).

**Table 3 plants-13-00632-t003:** Mortality of pupae and number of emerging IJs (mean ± SE) from *D. suzukii* pupae in combination with insecticides and *H. amazonensis* IBCBn 24 or *H. bacteriophora* HB.

Treatment	Pupal Mortality %	Number of IJs Emerged per Pupa
*H. amazonensis*	77.50 ± 3.13 b	297.62 ± 9.98 a
*H. bacteriophora*	78.75 ± 4.40 b	262.62 ± 7.76 b
Malathion	12.50 ± 3.13 c	-
Phosmet	6.25 ± 2.63 c	-
Azadirachtin	8.75 ± 2.26 c	-
Spinetoram	21.25 ± 2.26 c	-
Novaluron	12.50 ± 2.50 c	-
Malathion + *H. amazonensis*	81.25 ± 2.95 ab	87.62 ± 2.47 f
Phosmet + *H. amazonensis*	82.50 ± 5.26 ab	183.37 ± 2.52 d
Azadirachtin + *H. amazonensis*	87.50 ± 3.65 ab	124.00 ± 2.80 e
Spinetoram + *H. amazonensis*	95.00 ± 1.88 a	184.00 ± 4.15 d
Novaluron + *H. amazonensis*	88.75 ± 3.50 ab	27.62 ± 2.20 g
Malathion + *H. bacteriophora*	85.00 ± 3.27 ab	90.37 ± 1.23 f
Phosmet + *H. bacteriophora*	87.50 ± 2.50 ab	215.12 ± 3.35 c
Azadirachtin + *H. bacteriophora*	78.50 ± 2.26 b	120.87 ± 2.81 e
Spinetoram + *H. bacteriophora*	88.75 ± 3.50 ab	182.00 ± 1.87 d
Novaluron + *H. bacteriophora*	75.00 ± 1.88 b	44.87 ± 0.91 g
Control	7.50 ± 2.50 c	-
F	132.06	230.13
d.f	17.126	11.84
*p*	<0.0001	<0.0001

Different letters denote differences identified by the Tukey test, *p* < 0.05.

**Table 4 plants-13-00632-t004:** Insecticides used in bioassays to evaluate their compatibility with *S. brazilense* IBCBn 06, *S. carpocapsae* IBCBn 02, *H. amazonensis* IBCBn 24, and *H. bacteriophora* HB.

Active Ingredient	Trade Name	Registered Dose	Dose ^a^ (c.p)	Chemical Group	Mode of Action
**Deltamethrin**	Decis^®^ 25 EC ^vi^	40 mL/100 L^−1^	10.0	Pyrethroid	Sodium channel modulators
**Lambda-cyhalothrin**	Karate zeon^®^ 50 CS ^x^	4 g/100 L^−1^	1.0	Pyrethroid	Sodium channel modulators
**Malathion**	Malathion^®^ 1000 EC ^iv^	350 mL/100 L^−1^	87.5	Organophosphorus	Acetylcholinesterase enzyme inhibitors
**Phosmet**	Imidan^®^ 500 WP ^v^	200 g/100 L^−1^	50.0	Organophosphorus	Acetylcholinesterase enzyme inhibitors
**Azadirachtin**	Azamax^®^ 12 EC ^viii^	300 mL/100 L^−1^	12.5	Tetranotriterpenoid	Growth regulator
**Thiamethoxam**	Actara^®^ 250 WG ^iii^	10 g/100 L^−1^	2.5	Neonicotinoids	Acetylcholine agonist
**Acetamiprid**	Mospilan^®^ 725WG ^ii^	40 g/100 L^−1^	10.0	Neonicotinoid	Acetylcholine agonist
**Spinetoram**	Delegate^®^ 250WG ^vii^	20 mL/100 L^−1^	5.0	Spinosyn [5,18]	Acetylcholine receptor modulators
**Abamectin**	Vertimec^®^ 18 EC ^i^	70 mL/100 L^−1^	17.5	Avermectin	GABA agonists
**Novaluron**	Rimon 100 EC ^ix^	50 mL/100 L^−1^	12.5	Benzoylureas [12]	Chitin biosynthesis inhibitors

^vii^ Dow AgroSciences Industrial Ltd.a.n ^a^ Dose: g or mL of c.p. (commercial product)/100 L of water. Manufacturers (in São Paulo, SP, Brazil unless otherwise noted): ^ii^ Iharabras S/A Industriais Quimicas; ^i,iii,x^ Syngenta Proteção de Cultivos Ltd.a; ^viii^ UPL do Brasil Indústria e Comércio de Insumos Agropecuários S/A, Ituverava, Sao Paulo, SP, Brazil; ^vi^ Bayer S/A; ^iv^ FMC Química do Brasil Ltd.a; ^ix^ Adama Brasil S/A, Londrina, Paraná, PR, Brazil; ^v^ Cross Link Consultoria e Comércio Ltd.a, Barueri, São Paulo, SP, Brazil.

## Data Availability

The data is contained within the manuscript can be made available upon request.

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
