# Peer review of "Compatibility of Entomopathogenic Nematodes with Chemical Insecticides for the Control of Drosophila suzukii (Diptera: Drosophilidae)"

_plants, 2024, doi:10.3390/plants13050632_

Round 1

Reviewer 1 Report

Comments and Suggestions for Authors

The manuscript is a good work and written very well. The manuscript can be published in the Plants Journal. My comments and suggestions are given in the pdf attached.

Comments on the Quality of English Language

Moderate English editing is needed.

Author Response

Dear reviewer I accept all your suggestions. I am sending the comments in an attached file. Thank you very much for your contributions in improving
the quality of our manuscript.
Regards

Flavio R.M. Garcia

Reviewer 2 Report

Comments and Suggestions for Authors

The authors have focused on an important subject with very practical implications. It is quite interesting that although the concept of combining entomopathogens with synthetic or biological insecticides the application is really not that widespread. Maybe this work will give it more prominence.

Comments on the Quality of English Language

Some fixing (ie editorial work) is recommended. That would of course include English grammar.

Author Response

(The authors gave the same response as above.)

Reviewer 3 Report

Comments and Suggestions for Authors

Biological control with entomopathogenic nematodes (EPNs) and chemical insecticides suppress the spotted-wing drosophila, Drosophila suzukii, but the compatibility of the two approaches together requires further examination. Based on this issue, the authors carried out this experiment to evaluate the compatibility of four EPNs (IBCBn 06, IBCBn 02, IBCBn 24, and HB) with ten chemical insecticides for the control of D. suzukii. The topic is some interesting on biological control. While there are some shortcomings and errors, which were following as:

 Q1: Material and Methods: Not clear size of the used glass container just to say that 85 mm. 25 mm ± 0.5! Diameter, length or width? And Not clear about the photoperiod! Total 12 h per day or L:D=12 h: 12h?

 Q2: Table 2 and Table 3: How about the significant difference among the treatments of S. brazilense, S. carpocapsae, H. amazonensis and H. bacteriophor for same insectidice in a row?

 Q3: Table 3: Why was so low data (16.00) in contrast to other CK (38.00-42.00) for the control of S. brazilense IBCBn 06? Same control, while largely different data!

 Q4: Table 4: Mortality of D. suzukii pupae? For the respective treatments using the five insecticides solely without treated with EPNs, there are emerging IJs from D. suzukii pupae with 6.25% - 21.25% mortality? Confused results!

Author Response

(The authors gave the same response as above.)

Round 2

Reviewer 3 Report

Comments and Suggestions for Authors

This version has been revised based on the comments of the reviewers.

Suggest to accepting this MS.